# Recurrent networks of coupled Winner-Take-All oscillators for solving constraint satisfaction problems

**Hesham Mostafa, Lorenz K. Müller, and Giacomo Indiveri**
Institute for Neuroinformatics
University of Zurich and ETH Zurich
{hesham,lorenz,giacomo}@ini.uzh.ch

## Abstract

We present a recurrent neuronal network, modeled as a continuous-time dynamical system, that can solve constraint satisfaction problems. Discrete variables are represented by coupled Winner-Take-All (WTA) networks, and their values are encoded in localized patterns of oscillations that are learned by the recurrent weights in these networks. Constraints over the variables are encoded in the network connectivity. Although there are no sources of noise, the network can escape from local optima in its search for solutions that satisfy all constraints by modifying the effective network connectivity through oscillations. If there is no solution that satisfies all constraints, the network state changes in a seemingly random manner and its trajectory approximates a sampling procedure that selects a variable assignment with a probability that increases with the fraction of constraints satisfied by this assignment. External evidence, or input to the network, can force variables to specific values. When new inputs are applied, the network re-evaluates the entire set of variables in its search for states that satisfy the maximum number of constraints, while being consistent with the external input. Our results demonstrate that the proposed network architecture can perform a deterministic search for the optimal solution to problems with non-convex cost functions. The network is inspired by canonical microcircuit models of the cortex and suggests possible dynamical mechanisms to solve constraint satisfaction problems that can be present in biological networks, or implemented in neuromorphic electronic circuits.

## 1   Introduction

The brain is able to integrate noisy and partial information from both sensory inputs and internal states to construct a consistent interpretation of the actual state of the environment. Consistency among different interpretations is likely to be inferred according to an internal model constructed from prior experience [1]. If we assume that a consistent interpretation is specified by a proper configuration of discrete variables, then it is possible to build an internal model by providing a set of constraints on the configurations that these variables are allowed to take. Searching for consistent interpretations under this internal model is equivalent to solving a max-constraint satisfaction problem (max-CSP). In this paper, we propose a recurrent neural network architecture with cortically inspired connectivity that can represent such an internal model, and we show that the network dynamics solve max-CSPs by searching for the optimal variable assignment that satisfies the maximum number of constraints, while being consistent with external evidence.

Although there are many efficient algorithmic approaches to solving max-CSPs, it is still not clear how these algorithms can be implemented as biologically realistic dynamical systems. In particular, a challenging problem in systems whose dynamics embody a search for the optimal solution of a max-CSP is escaping from local optima. One possible approach is to formulate a stochastic neural network that samples from a probability distribution in which the correct solutions have higher

probability [2]. However, the stochastic network will continuously explore the solution space and will not stabilize at fully consistent solutions. Another possible solution is to use simulated annealing techniques [3]. Simulated annealing techniques, however, cannot be easily mapped to plausible biological neural circuits due to the cooling schedule used to control the exploratory aspect of the search process. An alternative deterministic dynamical systems approach for solving combinatorial optimization problems is to formulate a quadratic cost function for the problem and construct a Hopfield network whose Lyapunov function is this cost function [4]. Considerable parameter tuning is needed to get such networks to converge to good solutions and to avoid local optima [5]. The addition of noise [6] or the inclusion of an initial chaotic exploratory phase [7] in Hopfield networks partially mitigate the problem of getting stuck in local optima.

The recurrent neural network we propose does not need a noise source to carry out the search process. Its deterministic dynamics directly realize a form of "usable computation" [8] that is suitable for solving max-CSPs. The form of computation implemented is distributed and "executive-free" [9] in the sense that there is no central controller managing the dynamics or the flow of information. The network is cortically inspired as it is composed of coupled Winner-Take-All (WTA) circuits. The WTA circuit is a possible cortical circuit motif [10] as its dynamics can explain the amplification of genico-cortical inputs that was observed in intracellular recordings in cat visual cortex [11]. In addition to elucidating possible computational mechanisms in the brain, implementing "usable computation" with the dynamics of a neural network holds a number of advantages over conventional digital computation, including massive parallelism and fault tolerance. In particular, by following such dynamical systems approach, we can exploit the rich behavior of physical devices such as transistors to directly emulate these dynamics, and obtain more dense and power efficient computation [12]. For example, the network proposed could be implemented using low-power analog current-mode WTA circuits [13], or by appropriately coupling silicon neurons in neuromorphic Very Large Scale Integration (VLSI) chips [14].

In the next section we describe the architecture of the proposed network and the models that we use for the network elements. Section 3 contains simulation results showing how the proposed network architecture solves a number of max-CSPs with binary variables. We discuss the network dynamics in Section 4 and present our conclusions in Section 5.

## 2   Network Architecture

The basic building block of the proposed network is the WTA circuit in which multiple excitatory populations are competing through a common inhibitory population as shown in Fig. 1a. When the excitatory populations of the WTA network receive inputs of different amplitudes, their activity will increase and be amplified due to the recurrent excitatory connections. This will in turn activate the inhibitory population which will suppress activity in the excitatory populations until an equilibrium is reached. Typically, the excitatory population that receives the strongest external input is the only one that remains active (the network has selected a winner). By properly tuning the connection strengths, it is possible to configure the network so that it settles into a stable state of activity (or an attractor) that persists after input removal [15].

### 2.1   Neuronal and Synaptic Dynamics

The network that we propose is a population-level, rate-based network. Each population is modeled as a linear threshold unit (LTU) which has the following dynamics:

$$\tau_i \dot{x}_i(t) + x_i(t) = \max(0, \sum_j w_{ji}(t)x_j(t) - T_i) \qquad (1)$$

where $x_i(t)$ is the average firing rate in population $i$, $w_{ji}(t)$ is the connection weight from population $j$ to population $i$, and $\tau_i$ and $T_i$ are the time constant and the threshold of population i respectively. The steady state population activity in eq. 1 is a good approximation of the steady state average firing rate in a population of integrate and fire neurons receiving noisy, uncorrelated inputs [16]. For a step increase in mean input, the actual average firing rate in a population settles into a steady state after a number of transient modes have died out [17] but in eq. 1, we assume the firing rate approaches steady state only through first order dynamics.

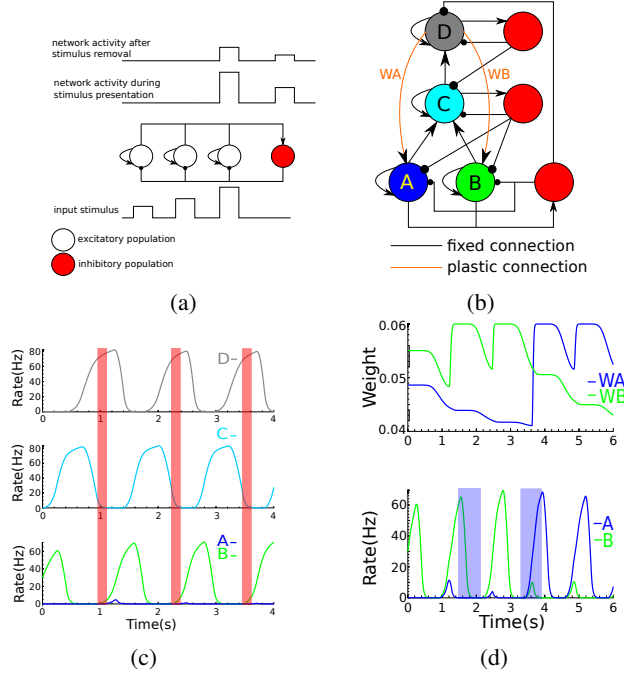

Figure 1: (a) A single WTA network. (b) Three coupled WTA circuits form the network representation of a single binary variable. Circles labeled A,B,C, and D are excitatory populations. Red circles on the right are inhibitory populations. (c) Simulation results of the network in (b) showing activity in the four excitatory populations. Shaded rectangles indicate the time intervals in which the state of the oscillator can be changed by external input. (d) Switching the state of the oscillator. The bottom plot shows the activity of the A and B populations. External input is applied to the A population in the time intervals denoted by the shaded rectangles. While the first input has no effect, the second input is applied at the right time and triggers a change in the variable/oscillator state. The top plot shows time evolution of the weights WA and WB.

The plastic connections in the proposed network obey a learning rule analogous to the Bienenstock-Cooper-Munro (BCM) rule [18]:

$$\dot{w}(t) = Ku(t)\left(\frac{(w(t) - w_{min})[v_{th} - v(t)]^-}{\tau_{dep}} + \frac{(w_{max} - w(t))[v(t) - v_{th}]^+}{\tau_{pot}}\right) \qquad (2)$$

where $[x]^+ = \max(0, x)$, and $[x]^- = \min(0, x)$. $w(t)$ is the connection weight, and $u(t)$ and $v(t)$ are the activities of the source and target populations respectively. The parameters $w_{min}$ and $w_{max}$ are soft bounds on the weight, $\tau_{dep}$ and $\tau_{pot}$ are the depression and potentiation time constants respectively, $v_{th}$ is a threshold on the activity of the target population that delimits the transition between potentiation and depression, and $K$ is a term that controls the overall speed of learning or the plasticity rate. The learning rule captures the dependence of potentiation and depression induction on the postsynaptic firing rate [19].

## 2.2 Variable Representation

Point attractor states in WTA networks like the one shown in Fig. 1a are computationally useful as they enable the network to disambiguate the inputs to the excitatory populations by making a categorical choice based on the relative strengths of these inputs. Point attractor dominated dynamics promote noise robustness at the expense of reduced input sensitivity: external input has to be large to move the network state out of the basin of attraction of one point attractor, and into the basin of attraction of another.

In this work, instead of using distinct point attractors to represent different variable values, we use limit cycle attractors. To obtain limit cycle attractors, we asymmetrically couple a number of WTA circuits to form a loop as shown in Fig. 1b. This has the effect of destroying the fixed point

attractors in each WTA stage. As a consequence, persistent activity can no longer appear in a single WTA stage if there is no input. If we apply a short input pulse to the bottom WTA stage of Fig. 1b, we start oscillatory activity and we observe the following sequence of events: (1) the activity in the bottom WTA stage ramps up due to recurrent excitation, and when it is high enough it begins activating the middle WTA stage; (2) activity in the middle WTA stage ramps up and as activity in the inhibitory population of this stage rises, it shuts down the bottom stage activity; activity in the middle WTA stage keeps on increasing until it activates the top stage; (3) activity in the top WTA stage increases, shuts down the middle stage, and provides input back into the bottom stage via the plastic connections. As a consequence, a bump of activity continuously jumps from one WTA stage to the next. Since the stages are connected in a loop, the network will exhibit oscillatory activity. There are two stable limit cycles that the network trajectory can follow. The limit cycle chosen by the network depends on the outcome of the winner selection process in the bottom WTA stage. The limit cycles are stable as the weak coupling between the stages leaves the signal restoration properties of the destroyed attractors intact allowing activity in each WTA stage to be restored to a point close to that of the destroyed attractor. The winner selection process takes place at the beginning of each oscillation period in the bottom WTA stage. In the absence of external input, the dynamics of the winner selection process in the bottom stage will favor the population that receives the stronger projection weight from D. These projection weights obey the plasticity rule given by eq. 2.

The oscillatory network in Fig. 1b can represent one binary variable whose value is encoded in the identity of the winning population in the bottom WTA stage, which determines the limit cycle the network follows. The identity of the winning population is a reflection of the relative strengths of WA and WB. More than two values can be encoded by increasing the number of excitatory populations in the bottom WTA stage. Fig. 1c shows the simulation results of the network in Fig. 1b when the weight WB is larger than WA. This is expressed by a limit cycle in which populations B,C, and D are periodically activated.

During the winner selection process in the bottom WTA stage, the WTA circuit is very sensitive to external input, which can bias the competition towards a particular limit cycle. Once the winner selection process is complete, i.e, activity in the winning population has ramped up to a high level, the WTA circuit is relatively insensitive to external input. This is illustrated in Fig. 1d, where input is applied in two different intervals. The first external input to population A arrives after the winner, B, has already been selected so it is ineffective. A second external input having the same strength and duration as the first input arrives during the winner selection phase and biases the competition towards A. As soon as A wins, the plasticity rule in eq. 2 causes WA to potentiate and WB to depress so that activity in the network continues to follow the new limit cycle even after the input is removed.

## 2.3 Constraint Representation

Each variable, as represented by the network in Fig. 1b, is a multi-stable oscillator. Pair-wise constraints can be implemented by coupling the excitatory populations of the bottom WTA stages of two variables. Fig 2a shows the implementation of a constraint that requires two variables to be unequal, i.e., one variable should oscillate in the cycle involving the A population, and the other in the cycle involving the B population. Variable $X1$ will maximally affect $X2$ when the activity peak in the bottom WTA stage of $X1$ coincides with the winner selection interval of $X2$ and vice versa. The coupling of the middle and top WTA stages of the two variables in Fig. 2a is not related to the constraint, but it is there to prevent coupled variables in large networks from phase locking. We explain why this is important in the next section. We define the zero phase point of a variable as the point at which activity in the winning excitatory population in the bottom WTA stage reaches a peak and we assume the phase changes linearly during an oscillation period (from one peak to the next). The phase difference between two coupled variables determines the direction and strength of mutual influence. This can be seen in Fig. 2b. Initially the constraint is violated as both variables are oscillating in the A cycle. $X1$ gradually begins to lead $X2$ until at a particular phase difference, input from $X1$ is able to bias the competition in $X2$ so that the B population in $X2$ wins even though the A population is receiving a stronger projection from the D population in $X2$.

A constraint involving more than two variables can be implemented by introducing an intermediate variable which will in general have a higher cardinality than the variables in the constraint (the cardinality of a variable is reflected in the number of excitatory populations in the bottom WTA stage; the middle and top WTA stages have the same structure irrespective of cardinality). An example is shown in Fig. 2c where three binary variables are related by an $XOR$ relation and the

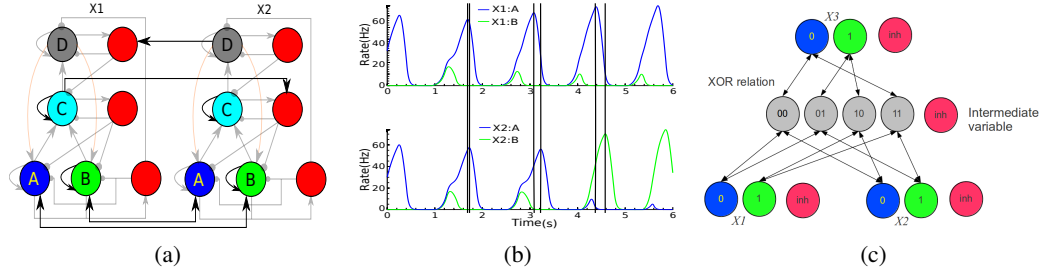

Figure 2: (a) Coupling $X1$ and $X2$ to implement the constraint $X1 \neq X2$. (b) Activity in the A and B populations of $X1$ and $X2$ that are coupled as shown in (a). (c) Constraint involving three variables: $X1\,XOR\,X2 = X3$. Only the bottom WTA stages of the four variables and the inter-variable connections coupling the bottom WTA stages are shown.

intermediate variable has four possible states. The tertiary $XOR$ constraint has been effectively broken down into three pair-wise constraints. The only states, or oscillatory modes, of $X1$, $X2$, and $X3$ that are stable under arbitrary phase relations with the intermediate variable are the states which satisfy the constraint $X1\,XOR\,X2 = X3$.

# 3 Solving max-CSPs

From simulations, we observe that the phase differences between the variables/oscillators are highly irregular in large networks comprised of many variables and constraints. These irregular phase relations enable the network to search for the optimal solution of a max-CSP. The weight attached to a constraint is an analogue quantity that is a function of the phase differences between the variables in the constraint. The phase differences also determine which of the variables in a violated constraint changes in order to satisfy the constraint (see Fig. 2b). The irregular phase relations result in a continuous perturbation of the strengths of the different constraints by modulating the effective network connectivity embodying these constraints. This is what allows the network to escape from the local optima of the underlying max-CSP. At a local optimum, the ongoing perturbation of constraint strengths will eventually lead to a configuration that de-emphasizes the currently satisfied constraints and emphasizes the unsatisfied constraints. The transiently dominant unsatisfied constraints will reassign the values of the variables in their domain and pull the network out of the local optimum. The network thus searches for optimal solutions by effectively perturbing the underlying max-CSP. Under this search scheme, states that satisfy all constraints are dynamically stable since any perturbation of the strengths of the constraints defining the max-CSP will result in a constraints configuration that reinforces the current fully consistent state of the network.

In principle, if some variables/oscillators phase-lock, then the weights of the constraint(s) among these variables will not change anymore, which will impact the ability of the network to find good solutions. In practice, however, we see that this happens only in very small networks, and not in large ones, such as the networks described in the following sections.

## 3.1 Network Behavior in the Presence of a Fully Consistent Variable Assignment

We simulated a recurrent neuronal network that represents a CSP that has ten binary variables and nine tertiary constraints (see Fig. 3a). Each variable is represented by the network in Fig. 1b. Each tertiary constraint is implemented by introducing an intermediate variable and using a coupling scheme similar to the one in Fig. 2c. We constructed the problem so that only two variable assignments are fully consistent. The problem is thus at the boundary between over-constrained and under-constrained problems which makes it difficult for a search algorithm to find the optimum [20].

We ran 1000 trials starting from random values for the synaptic weights within each variable (each variable effectively starts with a random value). The network always converges to one of the optimal variable assignments. Fig. 3b shows a histogram of the number of oscillation cycles needed to converge to an optimal solution in the 1000 trials. The number of cycles is averaged over the ten variables as the number of cycles needed to converge to an optimal solution is not the same for

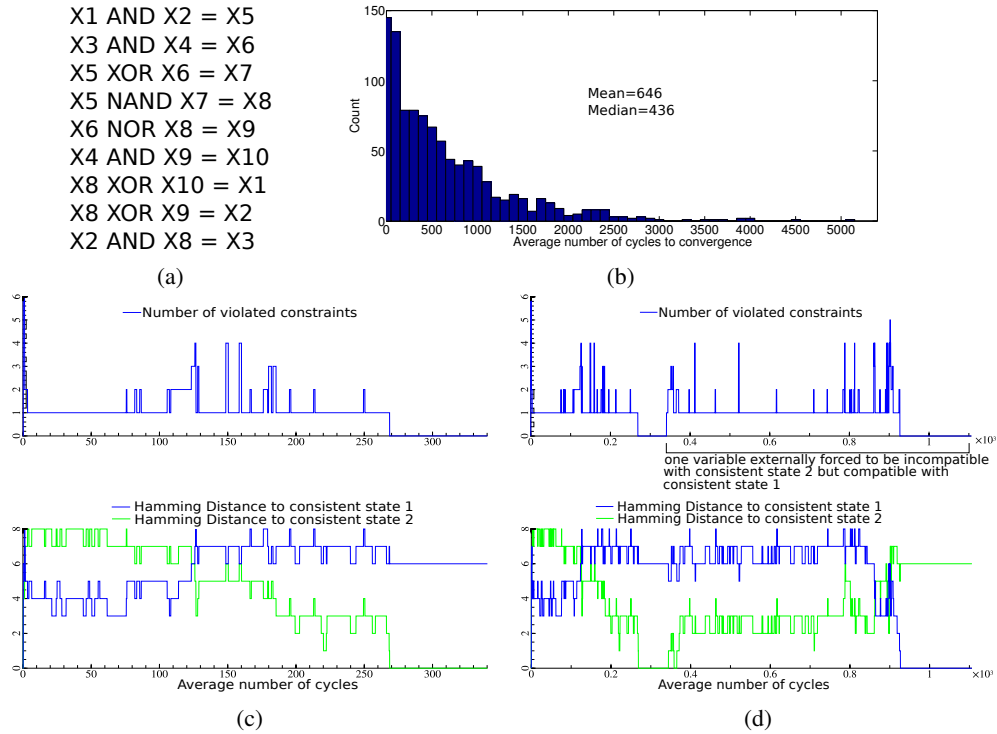

Figure 3: Solving a CSP with ten binary variables and nine tertiary constraints. (a) CSP definition. (b) Histogram of the number of cycles needed for convergence, averaged over all ten variables, in 1000 trials. (c) Evolution of network state in a sample trial. The top plot shows the number of constraints violated by the variable assignment decoded from the network state. The bottom plot shows the Hamming distance between the decoded variable assignment to each of the two fully consistent solutions. (d) One variable is externally forced to take a value that is incompatible with the current fully consistent variable assignment. The search resumes to find a fully consistent variable assignment that is compatible with the external input.

all variables. Although the sub-networks representing the variables are identical, each oscillates at a different instantaneous frequency due to the non-uniform coupling and switching dynamics. Fig. 3c shows how the network state evolves in a sample trial. Due to the continuous perturbation of the weights caused by the irregular phase relations between the variables/oscillators, the network sometimes takes steps that lead to the violation of more constraints. This prevents the network from getting stuck in local optima.

We model the arrival of external evidence by activating an additional variable/oscillator that has only one state, or limit cycle, and which is coupled to one of the original problem variables. External evidence in this case is sparse since it only affects one problem variable. External evidence also does not completely fix the value of that one problem variable, but rather, the single state "evidence variable" affects the problem variable only at particular phase differences between the two. Fig. 3d shows that the network is able to take the external evidence into account by searching for, and finally settling into, the only remaining fully consistent state that accommodates the external evidence.

## 3.2 Network Behavior in the Absence of Fully Consistent Variable Assignments

As shown in the previous section, if a fully consistent solution exists, the network state will end up in that solution and stay there. If no such solution exists, the network will never settle into one variable assignment, but will keep exploring possible assignments and will spend more time in solutions that satisfy more constraints. This behavior can be interpreted as a sampling process where each oscillation cycle lets one variable re-sample its current state; at any point in time, the network state represents a sample from a probability distribution defined over the space of all possible solutions

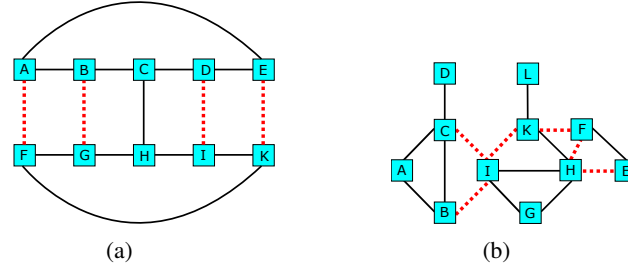

$$(a) \qquad\qquad\qquad (b)$$

Figure 4: Ising model type problems. Each square indicates a binary variable like in Fig. 1b; solid black lines denote a constraint requiring two variables to be equal, dashed red lines a constraint that requires two variables to be unequal. In both problems, all states violate at least one constraint.

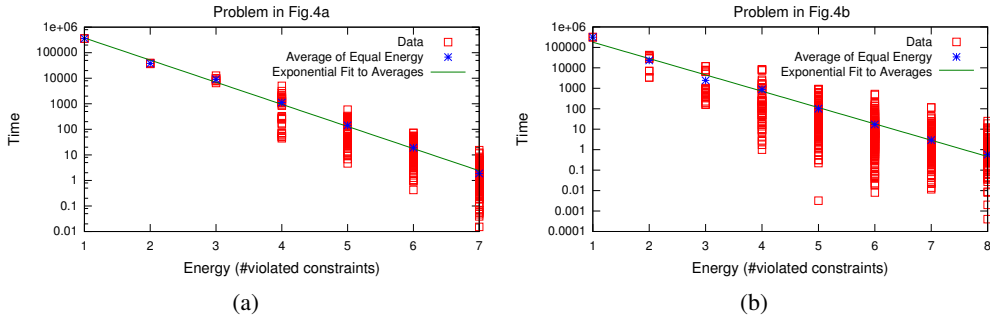

$$(a) \qquad\qquad\qquad\qquad (b)$$

Figure 5: Behavior of two networks representing the CSPs in Fig. 4. Red squares are data points (the time the network spent in one particular state), a blue star is the average time spent in states of equal energy and the green line is an exponential fit to the blue stars. (a) Note that at energies 1 and 2 there are two complementary states *each* that are visited almost equally often. (b) Not all assignments of energy 2 are equally probable in this case (not a finite samples artifact, but systematic) as can be seen in the bimodal distribution there. This is caused by variables that are part of only one constraint.

to the max-CSP, where more consistent solutions have higher probability. The oscillatory dynamics thus give rise to a decentralized, deterministic, and time-continuous sampling process. This sampling analogy is only valid when there are no fully consistent solutions. To illustrate this behavior, we consider two max-CSPs having an Ising-model like structure as shown in Figs. 4a, 4b. We describe the behavior of two networks that represent the max-CSPs embodied by these two graphs.

Let $E(s)$ be a function that maps a network state $s$ to the number of constraints it violates; this is analogous to an energy function and we will refer to $E(s)$ as the energy of state $s$. For the problem in Fig. 4a, we observe that the average time the network spends in states with energy $E$ is $t(E) = c_1 \exp(-c_2 E)$ as can be seen in Fig. 5a. The network spends almost equal times in complementary states that have low energy. Complementary states are maximally different but the network is able to traverse the space of intervening states, which can have higher energy, in order to visit the complementary states almost equally often.

We expect the network to spend less time in less consistent states; the higher the number of violated constraints, the more rapidly the variable values change because there are more possible phase relations that can emphasize a violated constraint. However, we do not have an analytical explanation for the good exponential fit to the energy-time spent relation. We expect a worse fit for high energies. For example, the network can never go into states where all constraints are violated even though they have finite energies.

For the problem in Fig. 4b, not all states of equally low energy are equally likely as can be seen in Fig. 5b. For example, the states of energy 2, where C and D (or K and L) are unequal, are less likely than other assignments of the same energy. This is not surprising. When C is in some state, D has no reason to be in a different state (no other variables try to force it to be different from C) apart from the memory in its plastic weights. We expect that this effect becomes small for sufficiently densely connected constraint graphs. The exponential fit to the averages is still very good in Fig. 5b.

# 4   Discussion

Oscillations are ubiquitous in cortex. Local field potential measurements as well as intracellular recordings point to a plethora of oscillatory dynamics operating in many distinct frequency bands [21]. One possible functional role for oscillatory activity is that it rhythmically modulates the sensitivity of neuronal circuits to external influence [22, 23]. Attending to a periodic stimulus has been shown to result in the entrainment of delta-band oscillations (2-4 Hz) so that intervals of high excitability coincide with relevant events in the stimulus [24]. We have used the idea of oscillatory modulation of sensitivity to construct multi-stable neural oscillators whose state, or limit cycle, can be changed by external inputs only in narrow, periodically recurring temporal windows. Selection between multiple limit cycles is done through competitive dynamics which are thought to underlie many cognitive processes such as decision making in prefrontal cortex [25].

External input to the network can be interpreted as an additional constraint that immediately affects the search for maximally consistent states. Continuous reformulation of the problem, by adding new constraints, is problematic for any approach that works by having an initial exploratory phase that slowly morphs into a greedy search for optimal solutions, as the exploratory phase has to be restarted after a change in the problem. For a biological system that has to deal with a continuously changing set of constraints, the search algorithm should not exhibit an exploratory/greedy behavior dichotomy. The search procedure used in the proposed networks does not exhibit this dichotomy. The search is driven solely by the violated constraints. This can be seen in the sampling-like behavior in Fig. 5 where the network spends less time in a state that violates more constraints.

The size of the proposed network grows linearly with the number of variables in the problem. CSPs are in general NP-complete, hence convergence time of networks embodying CSPs will grow exponentially (in the worst case) with the size of the problem. We observed that in addition to problem size, time to convergence/solution depends heavily on the density of solutions in the search space. We used the network to solve a graph coloring problem with 17 nodes and 4 colors (each oscillator/variable representing a node had 4 possible stable limit cycles). The problem was chosen so that there is an abundance of solutions. This led to a faster convergence to an optimal solution compared to the problem in Fig. 3a even though the graph coloring problem had a much larger search space.

# 5   Conclusions and Future Work

By combining two basic dynamical mechanisms observed in many brain areas, oscillation and competition, we constructed a recurrent neuronal network that can solve constraint satisfaction problems. The proposed network deterministically searches for optimal solutions by modulating the effective network connectivity through oscillations. This, in turn, perturbs the effective weights of the constraints. The network can take into account partial external evidence that constrains the values of some variables and extrapolate from this partial evidence to reach states that are maximally consistent with the external evidence and the internal constraints. For sample problems, we have shown empirically that the network searches for, and settles into, a state that satisfies all constraints if there is one, otherwise it explores the space of highly consistent states with a stronger bias towards states that satisfy more constraints. An analytic framework for understanding the search scheme employed by the network is a topic for future work.

The proposed network exploits its temporal dynamics and analog properties to solve a class of computationally intensive problems. The WTA modules making up the network can be efficiently implemented using neuromorphic VLSI circuits [26]. The results presented in this work encourage the design of neuromorphic circuits and components that implement the full network in order to solve constraint satisfaction problems in compact and ultra-low power VLSI systems.

## Acknowledgments

This work was supported by the European CHIST-ERA program, via the "Plasticity in NEUral Memristive Architectures" (PNEUMA) project and by the European Research council, via the "Neuromorphic Processors" (neuroP) project, under ERC grant number 257219.

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
