[Reviews · NeurIPS 2013]

Submitted by Assigned_Reviewer_5

Summary: The authors presented a recurrent neuronal network, modeled as a continuous-time dynamical system, that can solve constraint satisfaction problems.

Since the authors did not completely define their recurrent neuronal network, it is hard to understand at least for me. For example, v(t) and u(t) appear without definition in line 107. The single WTA network described in line 151 and Figure 1 (a) is not mathematically defined. Synaptic connections of the coupled WTA circuit described in lines 158-159 and Figure2 are not defined. The authors just said that they asymmetrically coupled a number of WTA circuits. I could not understand how the binary value is encoded as the state of the oscillatory network regarding lines 178-180. WA and WB appear without definition in line 191. The variables X1 and X2 appear without definition in line 199. Are X1 and X2 the synaptic weights or the names of two WTA circuits?

It is well known that neural network models tends to behave chaotic in the case that the synaptic connections dynamically change, for example, the short term synaptic depression. I suppose that that is the reason why the present model seems to be stochastic even thought it is defined as the deterministic system.

I also think that the system size of the present is too small (ten units with 9 constraint). For example, more than 10^3 units are usually discussed in the 3-SAT problem. I have to say that the example of the present paper is not enough in the framework of the CSP problem .
Summary: I could not understand contents of this paper, because the description seems to be incomplete as described in 1.

Submitted by Assigned_Reviewer_7

The authors propose a new approach for solving constraint satisfaction problems (CSPs) in a neural architecture. While previous attempts in this direction have largely relied on stochastic neuron models to prevent a neural network from getting stuck in local optima (e.g. through the use of Boltzmann machines), the proposed architecture is capable of finding optimal solutions to problems using purely deterministic network dynamics. This is achieved by coupling oscillator modules such that the strength of their mutual interaction depends on their phase difference. Although the paper does not provide a rigorous theoretical explanation for it, these phase differences are observed to vary irregularly in simulations, thus providing sufficient exploratory drive to escape local optima. In small simulations of a CSP with ten binary variables and nine tertiary constraints the network is observed to find either of the two correct solutions (which satisfy all constraints) in each trial. Interestingly, exploration is driven purely by violated constraints and thus, in contrast to stochastic approaches to solving CSPs, the network will stay in the global optimum once it is found (without the need for morphing into a greedy search as, for example, in simulated annealing). If no solution exists which satisfies all constraints the network behavior resembles a sampling process from a Gibbs distribution with a linear energy term. Unfortunately, a theoretical explanation of this behavior could not be provided by the authors.

The presented approach to solving CSPs in a deterministic neural architecture is, to the best of my knowledge, original and novel. It provides both an innovative approach for solving CSPs in neuromorphic hardware and a fresh perspective on oscillations and synchronization between oscillating modules (and lack thereof in the form of irregularly varying phase differences) in neuroscience.

The applications are non-trivial, although too few and too small to provide insight into the scalability of the method. On the positive side, the problems were chosen to be both illustrative and difficult in the sense that the energy landscape was designed to possess several local optima and more than one global optimum (especially in Fig. 3). On the negative side, the state spaces of the considered problems are very small (2^10 or 2^11). For such state spaces simple exhaustive search works well and is a much easier option. I would suspect that larger simulations would show that the algorithm still performs well, within what one may expect from a parallel iterative algorithm. Unfortunately the paper does not address scalability, and so the utility of the approach for larger problems remains unclear. At least a discussion of scalability is warranted and would be highly useful for the understanding of the method.

In terms of clarity and quality of figures and text the paper rates highly, yet no theoretical arguments were given for the observed behavior of the network. The paper is well organized, the model is for the most part clearly defined, the chosen examples are illustrative, the figures convey a clear message, and the explanations of the network dynamics are insightful. The lack of theoretical arguments is an obvious shortcoming of the paper. Furthermore, I'm missing a specification of the parameters used in the simulations (parameters in Eq. (1) and (2)). Were the same parameters used in the simulations for Fig 2, 3 & 4?

The paper is potentially significant, as it provides a new and relatively simple neural architecture for solving CSPs with deterministic neurons and has a large potential for neuromorphic applications. Furthermore, from the perspective of neuroscience, the potential relation between oscillations and the search for “consistent states” in the brain is intriguing and certainly worth pursuing.
Summary: This is an interesting and well written paper that provides an innovative model for solving CSPs in neural networks with oscillations, offering potential applications in neuromorphic engineering and a new perspective on oscillations in neuroscience. Two shortcomings of the paper are the lack of a theoretical treatment, and the lack of simulations concerning (or at least a discussion of) scalability.

Submitted by Assigned_Reviewer_8

Overview:

The paper studies a neuro-dynamic system that is able to solve
constraint satisfaction problems (CSP). The dynamic equations for
neural interaction and synaptic weight-changes are relatively
straight-forward (equations 1 and 2), the archictectures of the
networks are relatively complex. The elementary computational units
Are winner-take-all (WTA) type networks that oscillate.
It is stated that a crucial property of the elementary circuits is
their sensitivity to input only at specific points of their oscillatory
period(s). If the connectivity between such WTA units is chosen
to reflect a CSP, it is stated that the dynamics finally settles into
a state that represents a solution to the CSP. Simulations of the
neuro-dynamics for small and intermediately large CSPs empirically
show that the neuro-dynamics can find solutions, and that it can
find approximately optimal solutions if there are no fully consistent
variable assignments.


Evaluation:

The paper is generally well written and the main points become clear.
The question what computations neural circuits perform and how they are
performed is certainly important and interesting. There is also evidence
for WTA mechanisms that implement a form of possible competition between
neural untis. Also the effectiveness of input during certain points of
the phase is an interesting property because a small input differences
can be amplified (and response can be normalized). Even if weakly coupled
to an input, such networks can be very sensitive.

While the general setup is clear, I was missing a more extensive discussion
about the parameters of the network. Usually such parameters have to be tuned.
It does not become clear how difficult this is for the suggested system.
Also a discussion of related neuro-dynamic approaches is missing. Neural
dynamics that become sensitive to input in a critical period of their oscillation have been described before, sometimes in the form of sensitivity to initial
states. Such networks all have a WTA characteristics and at least some
should be discussed (see, e.g., work by O'Reilly, 2001; Spratling and
Johnson, 2002; Lucke, Malsburg, 2004). The difference to the here discussed
system is the goal: here it is finding solutions to CSPs, the other networks
address component finding in data. Finding CSP solutions is a well accepted
problem in computer science but it becomes much less clear why biological
networks should be optimized for such a problem - please discuss. Still it
is interesting from a technical point, and I agree that implementations of
inherently parallel neuro-dynamic approaches via analog VLSI are very
interesting. In this respect please discuss two questions, however: (1) isn't
the complexity of the problem mapped into the complexity of the network
architecture? (2) can one be confident that the problem solving dynamics do
not break down if the problem sizes are scaled up? What about the total number
of variables?

The behavior of the network is interesting including the behavior for not
fully satisfiable problems. The empirical finding of exponential scaling is
not uninteresting but the sampling analogy is taken too far for my taste.
The energy function could also be used for the satisfiable problem (Sec. 3.1).
But if the network remains in a solution there, it is not spending time
proportional to exp(-cE). Additionally, the authors themselves say that the
network will probably not explore high-energy states. So there is only a
coarse analogy here as far as I can see.

Also the criticism of other approaches is too general. Of course there are
many approaches on the technical side that have similar properties. Also it is
not clear how computationally costly the mechanisms to avoid local optima really
are for the studied dynamics - time to find optimal solutions
may scale exponentially with problem size - or the architecture does.


In response to author feedback:

The feedback has clarified most of my questions. I agree that scalability will
be exponential in the worst case. What would be interesting, though, is how
the average time to find a solution scales with problem size. This should be
a point for the revised discussion. I also assume that more care is taken regarding
the too broad criticism of other approaches, that it is discussed why CSP solving
is interesting for biological systems and about the analogy to sampling.

Given these improvements, I remain happy with my initial score.


Summary: Interesting neural dynamic approach to solve constraint satisfaction problems. May ultimately result in state-of-the-art approaches in terms of energy cost per CSP
problem solution if implemented on VLSI.

Submitted by Assigned_Reviewer_9

Summary: The authors propose a neural network model whose deterministic dynamics implement a search and ultimately global solution to a constraint satisfaction problem (CSP) when a solution exists. The fundamental computational unit of the network is a multi-stable oscillator that represent a binary variable. Multiple network modules are connected to represent multiple variables, and additional variables are defined as operations on these variables. When the modules are connected in such a way to implement a constraint (a desired operation on the variables), if the network dynamics are such that the constraint is violated, the the phase difference between different oscillators (variables) shifts such as to alter the certain state variables which cause the constraint violation. Once the relevant state variables have flipped, synaptic plasticity will cause this new state to become stable and persist, leading to a network state that is consistent with all of the desired constraints. When large simulations were conducted, the phase relationship between the oscillators was seen to vary irregularly, which resulted in sufficient exploratory drive for the network to escape local optima and search for solutions consistent with all of the desired constraints. With this property, the network has no need for often employed stochastic neuron models to escape local optima. Simulations were run showing that the network can find one of two globally optimal solutions to a 10 variable, 9 constraint problem.

Review: The paper is generally clear and well written. The figures are mostly well-chosen (although figure 4 doesn't seem to add much) and support the text. Network simulation examples are illustrative and provide the reader with insight as to how the network elements operate and how the oscillatory dynamics give rise to the plasticity which leads to a state that satisfies all the constraints. To the best of my knowledge, this approach is novel and the work posits a unique role for network oscillations. Furthermore, the model has the desirable property that, since search for a consistent solution is driven purely by violated constraints, external input can be evaluated by the network in real-time without the need for the system to switch between exploratory and greedy optimization modes. This makes the proposed model relevant to biological neural networks, which can rapidly assimilate external input with internal network state without global parameter alterations.

The largest shortcoming of the paper is that no explanation is provided as to why the network exhibits the exploratory phase varying that gives rise to its CSP solving ability for large network simulations. Considering the fact that the main result of the paper is a direct result of this network phenomenon some discussion of the origin of this phase variability would drastically improve the submission. Section 3.2 only weakly supports the main results of the paper and could be omitted to make room for a more elaborate discussion or analysis of the phase variability. Additionally, no mention is made as to the sensitivity of the network to parameters. A priori, most network models require careful parameter tuning so that the network dynamics are stable and exhibit a desired property. I imagine the same would be true for the proposed model, specifically to achieve the state of spontaneous phase shifting responsible for the network explorations. Alternatively, perhaps the rich, phase varying dynamics are relatively easy to achieve without careful parameter tuning due to the large size of the network, similar to a chaotic neural network state, in which case such a result would itself be interesting. Lastly, the scale of examples solved by the network- although useful to both illustrate how the network operates and to prove that it can solve CSPs- do not provide any insight into how the network's ability to solve CSPs scales with problem size. A more complete scaling analysis would be helpful to understand how the proposed network would perform on larger optimizations, or if the oscillatory dynamics would even perform the same function at a much larger scale.
Summary: This is an interesting, original model for solving CSPs in neural networks which employs a novel function for oscillatory dynamics. A rigorous analysis of the origin of the phase variability that gives rise to the network's ability to solve CSPs as well as the network's parameter sensitivity would greatly improve the paper.
Author Feedback

Author rebuttal: We thank the reviewers for their comments.
Regarding the issues raised by reviewer_5:
1) "v(t) and u(t) appear without definition in line 107."
v(t) and u(t) are defined in the text directly following the equation in
line 107, i.e. in lines 140 - 142

2) "The single WTA network described in line 151 and Figure 1 (a) is not mathematically defined."
The WTA network from Figure 1(a) is well described in [15], which is the source given for the paragraph in lines 82 - 90. The mathematical definition further follows directly from the definition of its components, given in equation (1). All synaptic weights within a single WTA are static

3)"Synaptic connections of the coupled WTA circuit described in lines 158-159 and Figure2 are not defined"
The synaptic connections in figure 2(a) follow the same notation as those in figure 1(b): Black lines indicate fixed weight connections. Orange lines indicate plastic connections that follow the plasticity rule given in equation (2). All synaptic connections are shown in Fig. 1b and Fig. 2a. In Fig. 2a, synaptic connections within the same variable/oscillator are shown in light colors to avoid cluttering the figure.

4)” I could NOT understand how the binary value is encoded”
The paragraph starting at line 177 describes how the value of the binary variable is encoded. We mention in several places that variable values are encoded in the identity of the limit cycle attractor chosen by the network representing the variable.

5) "WA and WB appear without definition in line 191"
WA and WB are introduced in figure 1(b) and (d) (ca. line 110) and equation (2). They are the strengths of two plastic weights. their effect is further described in lines 180 - 183.

6) "The variables X1 and X2 appear without definition in line 199"
In line 198 we refer the reader to figure 2(a) that illustrates X1 and X2. X1 and X2 are just generic variables and we mention in line 196 that a variable is represented by the network in Fig. 1b

Regarding the comments of the other three reviewers

7)”state spaces of the considered problems are very small”.
In the problem in Section 3.1, an intermediate variable with cardinality 4 was introduced
for each constraint. See Fig. 2c. The value of this intermediate variable can become inconsistent with the value of the 3 problem variables connected to it. There are thus 2^10*4^9 or 268 million possible states for the
variables/oscillators. This is the actual size of the space explored by the network

8)Scalability issues: CSPs are in general NP-hard. Having the variables in the problem update their values concurrently like in the proposed network does not change the fact that time to solution will grow (in the worst case) exponentially with problem size. The size of the network grows linearly.

9)”Lack of theoretical treatment”, “origin of phase variability”:
We recognize this as a shortcoming of the paper. A work similar to ours, [Hopfield and Tank, 1985], does not provide theoretical arguments for why the proposed Hopfield network is able to escape local optima to find good solutions to the travelling salesman problem, yet this work had significant impact. Many complex systems are analyzed using simulations, and we have provided a qualitative explanation for why the network behaves as it does. Phase variability is an emergent feature of the dynamics, similar to chaos. Like chaos in complex systems which in many cases is established using numerical simulations, irregularity in phase relations is a phenomenon that we observe in simulations. We are currently in the process of studying analytically the phase variability and plan to describe our preliminary results directly at NIPS and eventually in future publications.